# Identification of YWHAH as a Novel Brain-Derived Extracellular Vesicle Marker Post Long-Term Midazolam Exposure during Early Development

**DOI:** 10.3390/cells12060966

**Published:** 2023-03-22

**Authors:** Nghi M. Nguyen, Daniel Meyer, Luke Meyer, Subhash Chand, Sankarasubramanian Jagadesan, Maireen Miravite, Chittibabu Guda, Sowmya V. Yelamanchili, Gurudutt Pendyala

**Affiliations:** 1Department of Anesthesiology, University of Nebraska Medical Center (UNMC), Omaha, NE 68198, USA; 2Department of Genetics, Cell Biology, and Anatomy, University of Nebraska Medical Center (UNMC), Omaha, NE 68198, USA; 3Child Health Research Institute, University of Nebraska Medical Center (UNMC), Omaha, NE 68198, USA; 4National Strategic Research Institute, University of Nebraska Medical Center (UNMC), Omaha, NE 68198, USA

**Keywords:** 14-3-3 eta, Midazolam, NICU, extracellular vesicles, proteomics, neurodevelopment

## Abstract

Recently, the long-term use of sedative agents in the neonatal intensive care unit (NICU) has raised concerns about neurodevelopmental outcomes in exposed neonates. Midazolam (MDZ), a common neonatal sedative in the NICU, has been suggested to increase learning disturbances and cognitive impairment in children. However, molecular mechanisms contributing to such outcomes with long-term MDZ use during the early stages of life remain unclear. In this study, we for the first time elucidate the role of brain-derived extracellular vesicles (BDEVs), including mining the BDEV proteome post long-term MDZ exposure during early development. Employing our previously established rodent model system that mimics the exposure of MDZ in the NICU using an increasing dosage regimen, we isolated BDEVs from postnatal 21-days-old control and MDZ groups using a differential sucrose density gradient. BDEVs from the control and MDZ groups were then characterized using a ZetaView nanoparticle tracking analyzer and transmission electron microscopy analysis. Next, using RT-qPCR, we examined the expression of key ESCRT-related genes involved in EV biogenesis. Lastly, using quantitative mass spectrometry-based proteomics, we mined the BDEV protein cargo that revealed key differentially expressed proteins and associated molecular pathways to be altered post long-term MDZ exposure. Our study characterized the proteome in BDEV cargo from long-term MDZ exposure at early development. Importantly, we identified and validated the expression of YWHAH as a potential target for further characterization of its downstream mechanism and a potential biomarker for the early onset of neurodevelopment and neurodegenerative diseases. Overall, the present study demonstrated long-term exposure to MDZ at early development stages could influence BDEV protein cargo, which potentially impact neural functions and behavior at later stages of development.

## 1. Introduction

In the Neonatal Intensive Care Unit (NICU), it is common for neonates to undergo frequent and strenuous life-saving procedures [1]. While the limitation of these procedures is ideal for avoiding the use of analgesics and sedatives, this is not always possible. Early-stage neurodevelopment is a delicate and precise process, and as potent modifiers of signaling pathways essential to neurodevelopment, the use of these substances may potentially have adverse effects on the future neural health of NICU patients [2]. Therefore, the careful exploration of the long-term effects of these drugs is essential to determine which are safe for continued use. One of these commonly used sedatives is Midazolam (MDZ), a benzodiazepine frequently used to manage seizures and alleviate stress prior to surgery [3,4]. In previous studies, exposure to Midazolam has been linked to apoptotic neurodegeneration in the cerebral cortex and caudate nucleus and decreased hippocampal volume [1,5]. Furthermore, our lab has previously identified alterations in the synaptic proteome associated with long-term MDZ exposure, suggesting potential disturbances in neural plasticity and cytoskeletal architecture [3]. However, a comprehensive understanding of MDZ-induced effects on cellular composition and function is yet to be established. One crucial target for future studies is the brain-derived extracellular vesicle (BDEV). BDEV refers to a diverse group of membrane-bound nanovesicles, generally organized by origin and size as either exosomes, microvesicles, or apoptotic bodies [6]. Briefly, exosomes are phospholipid-bilayer bound vesicles ranging in size from about 50–100 nm and generated by exocytosis of multivesicular bodies. Microvesicles range from about 100–1000 nm and form via budding from the plasma membrane. Apoptotic bodies are 1–5 µm in diameter and are formed as blebs released from cells performing apoptosis [7]. BDEVs serve a wide variety of purposes, including conveying immune response [7,8], initiation of oncogenic cellular transformation [7,9], and lateral gene transfer [7,10], and contain diverse molecular cargo, including lipids, RNA, and protein [11]. As important facilitators of intercellular communication, cellular motility, cellular polarization, and even diseases including neurodegeneration and cancer, BDEVs represent a promising target for the study, as well as a therapeutic treatment, of MDZ-induced neural impairments [12,13]. For the first time, the current study utilized high-throughput mass spectrometry-based proteomics to identify differentially expressed proteins, as well as Gene Ontology (ClueGO) application to analyze our proteomic data and identify key differentially regulated molecular functions and biological processes associated with MDZ exposed BDEVs.

## 2. Materials and Methods

### 2.1. Animals

Pregnant Sprague–Dawley rats were obtained from Charles River Laboratories Inc. (Wilmington, MA, USA) and group-housed individually in a 12 h light-dark cycle. All animals were fed ad libitum and allowed to give birth naturally. All procedures and protocols were approved by the Institutional Animal Care and Use Committee (IACUC) of the University of Nebraska Medical Center (IACUC protocol: 19-130-01-EP) and conducted in accordance with the National Institutes of Health Guide for the Care and Use of Laboratory Animals.

### 2.2. Midazolam Treatment

The preclinical model system and the dosage regimen employed in this study have been established in our previously published work [3]. Briefly, at postnatal day (P) 3, rats were given subcutaneous injections of 1 mg/kg MDZ and ramped up to 10 mg/kg/day using the dose-escalation method until P21. Pups only received a single injection per day, following which they were placed beneath a heat lamp to maintain body temperature and were monitored for any distresses. Pups’ reflexes including posture, tail, cornea, and righting reflex were indicated as proper sedation. Brains were extracted at P21 1 h after the last treatment and stored at −80 °C for subsequent EV isolation.

### 2.3. BDEV Isolation

BDEVs were isolated using a sucrose density gradient as described in our previous studies [11,14,15]. Briefly, 450 mg of brain tissue was subjected to EV isolation. Brain tissue was homogenized and digested in 3.5 mL Hibernate A (HA) with 20 units/mL papain solution for 15 min at 37 °C. After 15 min of incubation in 37 °C, 6.5 mL HA containing antipain and protease inhibitor was added to the homogenized solution to stop the reaction. The solution was then centrifuged several times (at 300× *g* and 2000× *g* for 10 min, and 10,000× *g* for 30 min at 4 °C) to eliminate cellular debris and residual cells. The supernatant was then passed through a 0.22 µm syringe filter (Corning Incorporated (catalog # 431219), Corning, distributed by Thermo Scientific, Waltham, MA, USA). To maximize EV yield, samples were ultracentrifuged at 100,000× *g* for an hour at 4 °C, followed by purification using a density gradient separation technique. For the density gradient separation, sucrose concentrations ranging from 0.25 to 2 M were used, and the concentrated EVs pellet was resuspended in the 0.95 M layer and ultra-centrifuged at 200,000× *g* for 16 h at 4 °C. To obtain the purified EV pellet, fractions enhanced with EVs were subjected to additional ultracentrifugation at 100,000× *g*. These EV pellets were then resuspended in particle-free phosphate-buffered saline (PBS), and their protein content was determined using a BCA Protein Assay Kit (Thermo Scientific, Waltham, MA, USA) per the manufacturer’s manual.

### 2.4. BDEV Markers Validation

To establish the purity of the BDEVs, we examined the expression of positive markers: Hsp70 (SAB4200714, Sigma-Aldrich, St. Louis, MO, USA), Flot-1 (abcam41927, Abcam, Cambridge, UK), CD81 (MCA1846, Bio-Rad Laboratories, Inc., Hercules, CA, USA), CD63 (BD551458, BD Biosciences, Franklin Lakes, NJ, USA), HSP90 (C45G5, Cell Signaling Technology, DANVERS, MA, USA), and negative marker GM130 (BD610822, BD Biosciences, Franklin Lakes, NJ, USA), as described in previous papers [11,14,15]. Briefly, 25–30 μg of BDEV was subjected to Western blot under reducing (Hsp70, Flot-1, GM130) and non-reducing (CD81, CD63, Hsp90) conditions using 4–12% Bis-tris gels (Invitrogen, Waltham, MA, USA).

For YWHAH validation, 15 μg of BDEV was subjected to Western blot under reducing conditions using 10% Bis-tris gel (Invitrogen, Waltham, MA, USA) against YWHAH (15222-1-AP, Proteintech, Rosemont, IL, USA). Primary and secondary antibody dilutions were carried out according to the manufacturer’s suggestions. Blots were developed using Azure CSeries Imager (Azure Biosystems, Dublin, CA, USA) with SuperSignal West Pico Chemiluminescent Substrate (Thermo Scientific, Waltham, MA, USA).

### 2.5. TEM acquisition and Analysis

Transmission electron microscopy (TEM) on the BDEVs from saline and MDZ samples was performed as described in our previous studies [11,14,15]. Post-isolation, BDEV pellets were briefly suspended in 1X particle-free PBS. Then, 10 µL of the BDEV suspension was combined with 90 µL of the TEM fix buffer (2% glutaraldehyde, 2% paraformaldehyde, and 0.1 M phosphate buffer). Next, 10 µL of the sample was added onto a mesh copper grid for 2 min and subsequently drawn up using a piece of filter paper. The left-over film was allowed to dry for 2 min. NanoVan negative stain was later added onto the grid, drawn up via filter paper, and allowed to dry before imaging on a Tecnai G2 Transmission Electron Microscope (FEI, Hillsboro, OR, USA) that operated at 80 kV.

### 2.6. Zeta View Analysis

To comprehensively analyze the EVs’ size distributions and particle concentration information, we utilized ZetaView nanoparticle tracking analyzer PMX120 (Particle Metrix, Inning am Ammersee, Germany) along with the software ZetaView 8.05.12 SP2 as previously described [16]. The instrument was calibrated using 100 nm polystyrene nano standard particles prior to the actual analysis. A total of 10 μL of the sample was diluted to 1:10,000–1:30,000 in particle-free 1X PBS before loading to the instrument. Once samples were loaded, cell quality and particle drift were checked before capturing the video. The video was captured at a sensitivity of 65, a shuttle speed of 100, and a frame rate of 30 frames/s. The sample’s size (in nm) and concentration (in particle/mL) were determined after being read at 11 positions for three cycles.

### 2.7. RNA Extraction and RT-qPCR

Total RNA from the brain cortex was isolated from the randomly selected P21 pups (n = 12) from each treatment group at P21 using the Direct-Zol RNA kit (Zymo Research, Irvine, CA, USA) based on the manufacturer’s protocol. The total amount of extracted RNA was quantified by Epoch2 (BioTek Instruments, Winooski, VT, USA). RNA was then converted to cDNA using the Superscript IV kit (Invitrogen, Waltham, MA, USA). The expressions of ESCRT complexes (ESCRT-0, -I, -II, and -III) were detected by qPCR using a pre-designed TaqMan^TM^ Array 96-Well FAST Plate (REF#: 4413259, appliedbiosystems, Pleasanton, CA, USA) and QuantStudio™ 7 Flex Real-Time PCR System (cat: 4485701, appliedbiosystems, Pleasanton CA, USA). The TaqMan Plate consisted of 18S and GAPDH as housekeeping genes. Relative fold change was calculated based on the 2^-ddCT method. The list of TaqMan probes and the overall plate design are included in Appendix A.

### 2.8. Proteomics Analysis

Protein quantification was performed using Pierce BCA protein assay (Thermo Scientific, Rockford, IL, USA) and followed the procedure described in our earlier studies [3,17]. 50 µg of protein per sample from 6 animals was subjected to mass spectrometry (MS) analysis. MS analysis was established and managed with a UNMC Mass Spectrometry Core (Omaha, NE, USA). The analysis was based on the label-free quantitative mass spectrometry protocol. The details of the MS protocol and reagents used are described in our recently published studies [3,17].

The in-house mascot 2.6.2. (Matrix Science, Boston, MA, USA) search engine was used further to identify the proteins from tandem mass spectrometry data, as described in our published studies [3,17]. The search targeted total tryptic peptides and allowed for two missed cleavage sites. Carbamidomethylation of cysteine was set as a fixed modification. Acetylation of protein N-terminus and oxidized methionine were included as variable modifications. The highest permitted fragment mass error was 0.02 Da, and the precursor mass tolerance threshold was set at 10 ppm. A false discovery rate (FDR) of ≤1% was used to calculate the significance threshold of the ion score. Qualitative analysis was performed using progenesis QI proteomics 4.1 (Nonlinear Dynamics, Milford, MA, USA). 

### 2.9. Bioinformatics Analysis

Proteins were established as differentially expressed utilizing criteria of a *t*-test *p*-value < 0.05, a minimum of 2 unique peptides, and an absolute fold-change ≥ 1.5. Utilizing the Cytoscape plug-in ClueGO [18], gene ontology analysis was performed on the differentially expressed proteins (DEPs). Specifically, molecular functions and biological processes were targeted for GO enrichment analysis. A heatmap of all DEPs was created by employing function heatmap.2 in R (version 3.6.0) package, *gplots* [19].

### 2.10. Statistical Analysis

Post normalization, an unpaired *t*-test was conducted to detect significant DEPs following exposure to MDZ. Proteins with *p*-value < 0.05, ≥ 2 unique peptides, and an absolute fold-change ≥ 1.5 were considered significant. The downregulation of YWHAH was established as significant, utilizing a student’s unpaired *t*-test following Welch’s correction with *p* < 0.05. All statistical analysis was conducted using GraphPad Prism version 9.4 (LA Jolla, CA, USA).

## 3. Results

### 3.1. Repetitive Exposure of MDZ during Early Development Influences BDEV Size in the Brain

To elucidate whether long-term MDZ exposure impacts BDEV composition and subsequent protein cargo, we isolated BDEV from both saline (control) and MDZ (experimental) groups utilizing the ultracentrifugation method from the multiple sucrose gradients, as described in our previous studies [11,14,15]. We then evaluated the purity of isolated BDEV by means of Western blot analysis against positive and negative EV markers, which were established by the International Society for Extracellular Vesicles [20]. Figure 1A shows the presence of the positive markers Hsp70, Hsp90, Flotillin-1, CD63, and CD81 in the BDEVs and the absence of EVs in the negative marker GM130. ZetaView analysis showed a slight increase in the number of BDEVs released in post long-term MDZ exposure animals; however, the overall results were not significant (Figure 1B). All original blots containing all replicates are shown in Appendix A.

TEM analysis showed that BDEV particles had sphere-like shapes, and the most abundant size of the vesicles was around 200–300 nm in diameter (Figure 2). The BDEV size was not heavily influenced by long-term exposure to MDZ. Despite the insignificance of the results, smaller BDEV (<50 nm) was lesser abundant in the MDZ exposure (*p* = 0.063).

### 3.2. Chronic MDZ Exposure during Early Development Does Not Impact BDEV Biogenesis

Based on the slight increase in particle concentration and relatively larger BDEV particles in the MDZ-exposed groups, we next evaluated if early long-term exposure to MDZ impacts EV biogenesis machinery. Accordingly, using RT-qPCR, we looked at the expression levels of key Endosomal Sorting Complex Required for Transport (ESCRT) genes, which regulate EV biogenesis. As seen in Figure 3, we did not observe any significant changes in the different genes associated with the ESCRT complexes, suggesting that long-term exposure to MDZ during early development does not impact BDEV biogenesis.

### 3.3. Long-Term MDZ Exposure Alters the Proteome in EV Cargo

Whilst no significant changes were observed in the BDEV biogenesis, we next tested whether long-term MDZ exposure during the early stages of life induced alterations in the BDEV proteome. Accordingly, we subjected P21-purified BDEV from the control and MDZ groups to high throughput quantitative mass-spectrometry-based proteomics. A total of 3328 proteins were identified (Appendix A). Further employing a criterion of 2+ unique peptides, and *p* < 0.05, we identified 15 proteins to be differentially expressed between the two groups (Table 1). The heatmap (Figure 4) highlighted the differentially expressed proteins (DEPs) after MDZ exposure based on a criterion of 1.5 absolute fold-change and *p* < 0.05. A total of eight proteins were up-regulated, while seven were down-regulated with respect to the control group.

### 3.4. Gene Ontology Analysis of BDEV Reveals Potential Processes and Functions Associated with Long-Term MDZ Exposure

We next analyzed the biological processes and molecular functions enriched with these DEPs (Figure 5 and Figure 6) using the bioinformatics tool ClueGO analysis. The most abundant biological process was the corticosteroid receptor signaling pathway, with 30.0% of gene ontology (GO) terms associated with this process, followed by olfactory learning, regulation of adenylate activity, integrin activation, and the rest of the processes reported in Figure 5A, which all shared 10% of the GO terms. For the molecular function, 28.57% of DEPs are associated with the sodium channel activity and adenylate cyclase activity, while other molecular functions include ATP-dependent peptidase activity, LIM domain binding, and Inositol 1,4,5 trisphosphate binding, which all account equally for 14.29% of GO terms (Figure 6A). Figure 5B and Figure 6B represent all the possible connections and networks of those DEPs associated with GO terms and groups of functions or processes. A complete list of GO terms and associated genes can be found in Appendix A (biological processes) and Appendix A (molecular functions).

### 3.5. Identification of YWHAH as a Novel BDEV Marker with Long-Term MDZ Exposure during Early Development

The high-throughput proteomics generated several possible hits, and further validation was necessary. One hit we successfully identified and validated was 14-3-3 eta protein (YWHAH), an adapter protein responsible for intracellular signaling and regulatory pathways. YWHAH was down-regulated −3.27-fold in the MDZ-treated group (Table 1), and its protein expression change was further validated with Western Blot (Figure 7). All original blots containing all replicates are shown in Appendix A.

## 4. Discussion

The current study for the first time aims to decode how chronic long-term exposure to the sedative benzodiazepine MDZ during the early stages of development impacts the overall BDEV function. The main findings from our study underlined the up- and down-regulated differentially expressed proteins (DEPs) in the BDEV cargo that are potentially involved in various molecular functions (e.g., sodium channel regulatory, adenylate cyclase, and ATP-dependent peptidase activities) and biological processes (e.g., corticosteroid receptor signaling pathway, regulation of adenylate activity, and positive regulation of protein depolymerization). The results from our study provide new aspects concerning long-term MDZ exposure during early development, potentially affecting intercellular neural signaling and overall molecular functions.

In our study, we did not observe a significant difference in the size distribution of EVs, which implied that there is no difference in the EV subpopulations in the brain post MDZ exposure. However, we observed an interesting trend regarding the number of EVs released between saline and MDZ-exposed brains. Specifically, the total amount of particles produced by the MDZ group is slightly higher and more abundant at a larger size than the control (Figure 1B and Figure 2B). This suggests alterations in the inner EV cargo materials that affect EV size. Since we observed a heterogenous subpopulation of EVs in the brain, we further characterized the DEPs on the EV cargo.

In the developing brain, regular EV release contributes to normal brain development since EVs play an essential role in cell-to-cell communication in the central nervous system (CNS) [21]. The CNS structure, function, and homeostasis depend on cell-to-cell communication throughout the lifespan [22]. Notably, materials such as proteins, microRNA, lipids, etc., packed in the EV cargo could be characterized into biomarkers. Neurogenesis, gliogenesis, synaptogenesis, and myelination, are major CNS processes that start at the beginning of life and are critical for assuring the proper development in the brain to proceed to adulthood [23]. Therefore, any internal or external insults to the brain, such as exposure to neurotoxicity substances, during these critical processes will alter the brain’s homeostasis and make it more vulnerable to normal development. Studies established by Fauré et al. and Lachenal et al. implied that EVs could act as disposal mechanisms in synapses, where lysosomes are absent, to discard synaptic receptors as a response to the alterations of synaptic plasticity [24,25]. Notably, in many preclinical and clinical studies, exposure to MDZ during early developmental periods has been associated with alteration of synaptic plasticity subsequently leading to impaired cognitive function and an increased risk of developing learning disorders [26,27,28,29].

Interestingly, this study identified a connection between DEPs and potential networks associated with synaptic and disposable functions in the cell. Our network analysis based on ClueGO biological processes showed the association between E3 ubiquitin-protein ligase (NEDD4) and lysosome-associated membrane glycoprotein (LAMP2) proteins in targeting the vacuole (Figure 5), which is an organelle that helps sequester waste product in animal cells [30]. NEDD4 and LAMP2 are up-regulated in our Midazolam-exposed BDEVs (Figure 3 and Table 1). Importantly, in the CNS, up-regulation of NEDD4 has been noted in several neurodegenerative diseases such as Alzheimer’s, Parkinson’s disease, and Huntington’s disease [31,32]. Furthermore, Kwak et al.’s 2012 study pointed out that the increase in transcriptional activation of NEDD4 results from the escalation in neurotoxins and oxidative stress present in the brain [31]. On the other hand, the functional role of LAMP2 is maintaining lysosomal stability and actively participating in autophagy [33]. In neuropathology, LAMP2 concentration has been reported to be decreased in the patient’s cerebrospinal fluid (CSF) of Parkinson’s disease [34,35] and increased in patients with Alzheimer’s [36,37]. Further research would need to invest in determining the mechanism of why these proteins are packed more into the EV cargo, whether it would be a protection mechanism of the brain to cope with the repetitive exposure to MDZ, or if it would be a sort of biomarker that acts as an early onset signal for a developing neuropathology.

Our ClueGO analysis (Figure 5 and Figure 6) also revealed that the molecular and biological processes are mainly associated with sodium channel activity. One of the proteins related to this group that we identified was protein 14-3-3 eta or YWHAH. The 14-3-3 proteins, also known as YWHAx, where the “x” is a representation for isoforms including β, γ, ε, ζ, η, θ, and σ that are encoded by seven genes (*YWHAB/G/E/Z/H/Q/S*, respectively), is a highly conserved protein. YWHAx proteins are involved in many crucial cellular processes such as metabolism, protein trafficking, signal transduction, apoptosis, cell cycle regulation, transcription, stress responses, and malignant transformation [38]. With 14-3-3 proteins having a wide range of cellular and molecular roles and functions, it’s no surprise that this family is linked to various human disorders [39]. Some human diseases associated with 14-3-3 protein dysfunction are cardiomyopathy, cancer, and even hair pigmentation [39]. Notably, the YWHAx family is highly conserved and abundant in the brain, accounting for 1% soluble protein [40]. Since YWHAx proteins are the most abundant in the brain and are frequently present in cerebrospinal fluid in neurodegenerative illnesses, they may play a crucial role in the physiology and death of neurons [41]. In recent decades, the importance of the YWHAx family has been highlighted in various neurological and neuropsychiatric disorders, including schizophrenia, bipolar disorders, Parkinson’s disease, Alzheimer’s disease, and Creutzfeldt–Jakob disease [41,42].

Intriguingly, over the past decade, evidence has emerged that the 14-3-3 family is essential for the development of the nervous system, and especially cortical development. More recently, the 14-3-3 family has been recognized as a crucial regulator in several neurodevelopmental processes. Our results revealed that YWHAH was significantly down-regulated in proteomics data and Western blot (Figure 4 and Figure 7). A study by Kwon et al. demonstrated that ischemia injury decreased YWHAH expression in the hippocampus sub-regions. Moreover, the expression of YWHAH was also found to be decreased in kainic acid-induced neurotoxicity [43]. Another study conducted by Buret et al. pointed out that mice with YWHAH deficiency led to the impairment of outer and inner hair cells, and variants of YWHAH would lead to mitochondria fragmentation, making the cells more susceptible to apoptosis [41]. Noteworthy, these studies established the idea that the downregulation, imbalance, or dysfunction of YWHAH could be highly associated with subsequent neuronal loss, which eventually contributes to the development of neurodegenerative diseases. Evidence shows that more neuronal cell death occurs when exposed to MDZ in many *in vitro* and *in vivo* studies [5,28,44]. Thus, the downregulation of YWHAH post-MDZ exposure could possibly be implicated in neuronal cell death.

## 5. Conclusions

Our study for the first time performed a comprehensive analysis on how long term MDZ exposure at early development can brain function. Notably, using state of the art high throughput technologies, we mined the BDEV proteome and identified including validating the expression of YWHAH as a potential target. Future studies are aimed at further characterization of its downstream mechanisms both *in vitro* and *in vivo* and thus could pave way to develop it as a potential therapeutic to mitigate altered brain development with long term MDZ exposure in neonates. 

## Figures and Tables

**Figure 1 cells-12-00966-f001:**
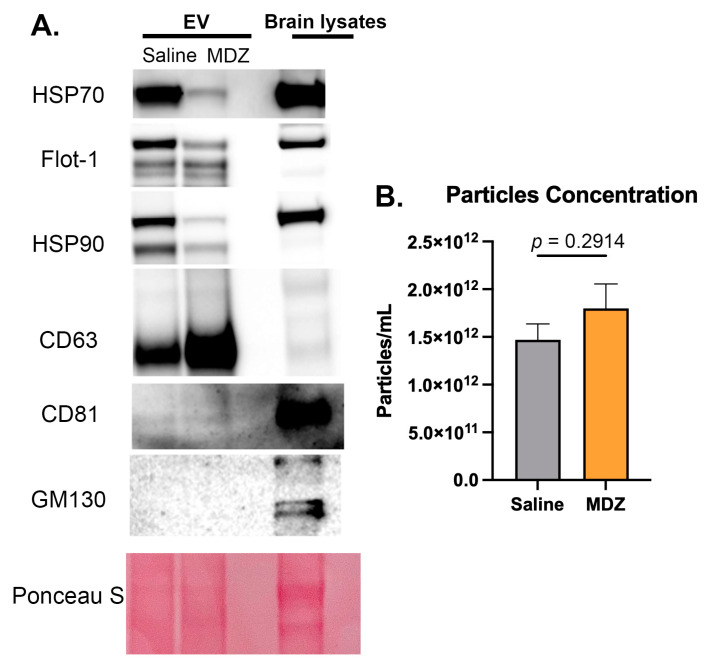
**Validation of BDEV purity.** (**A**) Western blot analysis on BDEV isolated from a control saline animal shows the expression of the positive markers Hsp70, Flot-1, Hsp90, CD63, and CD81. The negative marker GM130 was enriched in the whole brain lysate but absent in the EV fraction. Equal loading was confirmed via Ponceau S stain. (**B**) ZetaView nanoparticle tracking analysis shows a slight increase in BDEV particle concentration in rats exposed to MDZ (n = 18/group). However, there were no significant changes in the concentrations of isolated BDEVs from the two groups as determined by an unpaired *t*-test following Welch’s correction. Data are represented as Mean ± SEM.

**Figure 2 cells-12-00966-f002:**
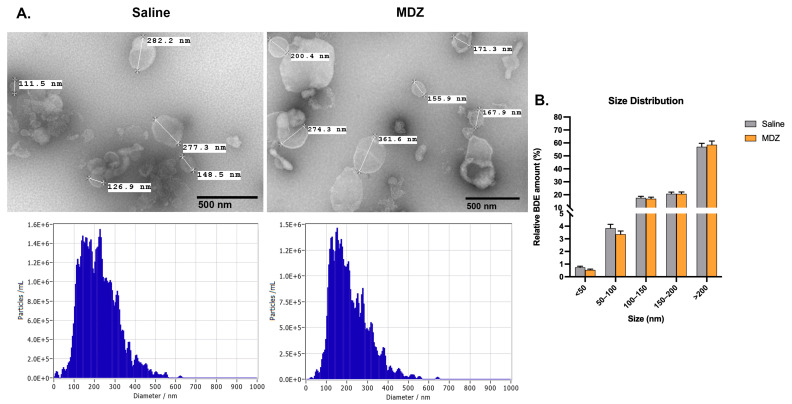
**Characterization of BDEV.** (**A**) BDEV isolated using a sucrose density gradient from the two groups was characterized using TEM (top), and Zeta-view analysis revealed no change in the sizes of particles with/without MDZ exposure (bottom). (**B**) Zeta-view analysis shows the different size distributions. Each group’s average BDEV sizes (in nm) are represented as Mean ± SEM (n = 18/group). Statistical significance is determined by multiple *t*-tests followed by Holm–Sidak correction.

**Figure 3 cells-12-00966-f003:**
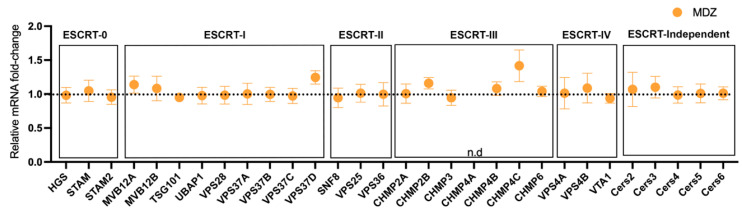
Long-term MDZ exposure during early development did not alter the biogenesis and the expression of ESCRT-dependent and independent pathway proteins. A custom qRT-PCR panel for EV biogenesis genes illustrated that genes involved in the ESCRT dependent (complexes 0, I, II, and III) and independent pathways did not alter upon exposure to MDZ (orange dot). Dotted line represents a baseline for Saline fold-change expression. Data represented as Mean ± SEM, n = 12/group, analyzed with a multiple *t*-test followed by Holm–Sidak correction. n.d = no data, no amplification observed.

**Figure 4 cells-12-00966-f004:**
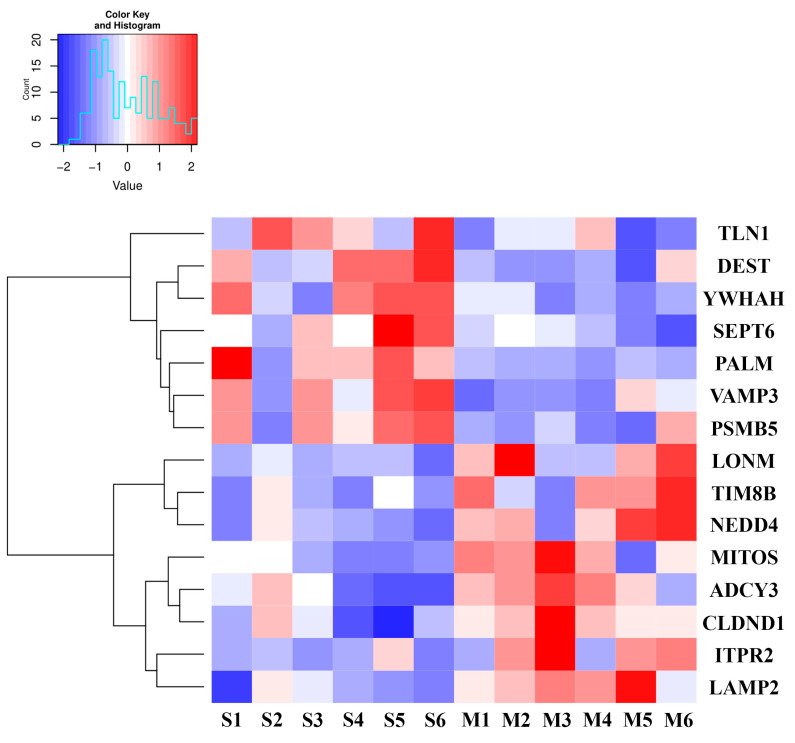
**Differential expression of BDEV proteins in male and female rats post MDZ exposure**. Heatmap showing the DEPs between the Midazolam and saline. S—Saline, M—Midazolam. Data were obtained from all six biological replicates from each group.

**Figure 5 cells-12-00966-f005:**
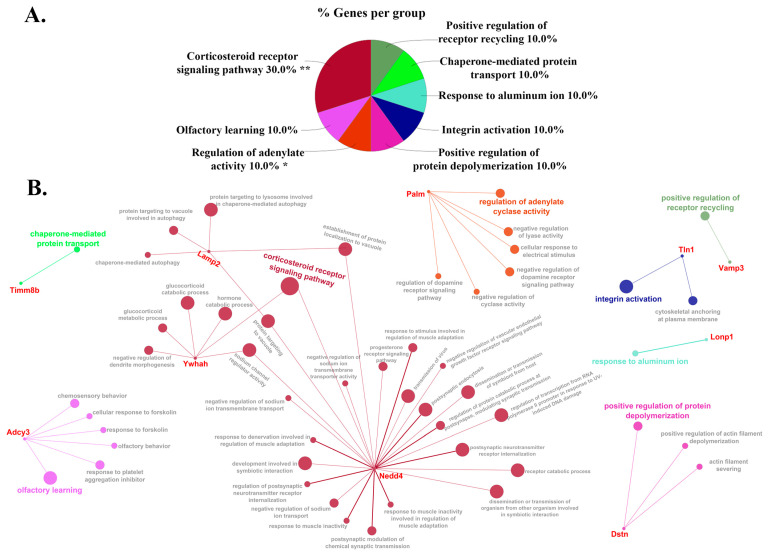
(**A**) Percentage of genes per group mapped to biological processes and (**B**) networks shows the interaction of proteins and their biological process for the DEPs using ClueGO. The asterisks represent the group term *p*-value representing each category. * *p* < 0.05 and ** *p* < 0.01.

**Figure 6 cells-12-00966-f006:**
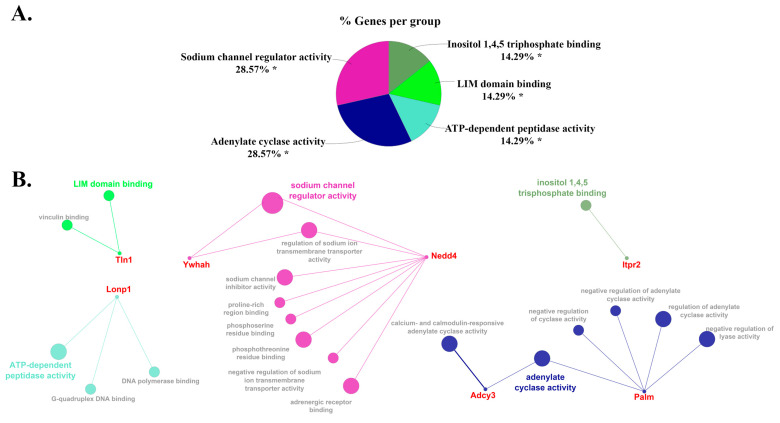
(**A**) Percentage of genes per group mapped to molecular functions and (**B**) networks shows the interaction of proteins and their molecular function for the DEPs using ClueGO. The asterisks represent the group term *p*-value representing each category. * *p* < 0.05.

**Figure 7 cells-12-00966-f007:**
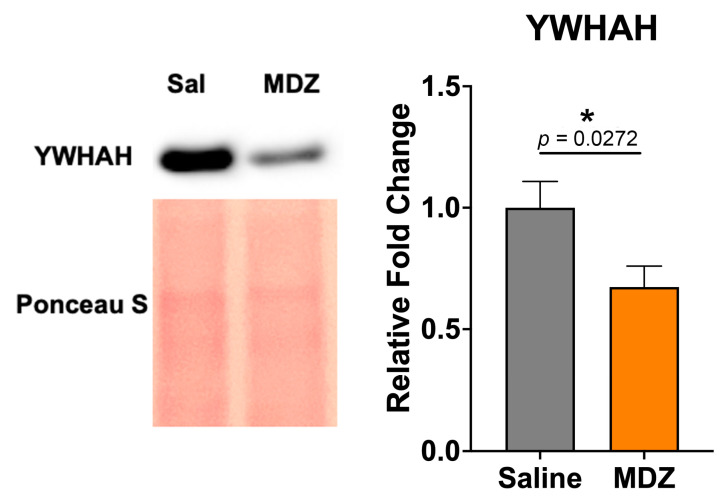
**Validation of YWHAH after MDZ exposure.** A representative Western Blot is depicted. Data is represented as Mean ± SEM (n = 13/group), and significance was determined with an unpaired *t*-test after Welch’s correction. * *p* = 0.0272.

**Table 1 cells-12-00966-t001:** **Significant DEPs in P21 BDEVs after MDZ exposure.** A criterion of 2+ unique peptides and an absolute fold change of >1.5 were used to select the potential hits. (+) and (−) sign represents up- and down-regulation, respectively, with respect to Saline samples.

Accession Number	Description (Protein Name)	Fold-Change
P62078	Mitochondrial import inner membrane translocase subunit (TIMM8B)	+3.86
Q9245D5	Lon protease homolog, mitochondrial (LONP1)	+2.99
Q5RKI8	Mitochondrial potassium channel ATP-binding subunit (ABCBC8)	+2.65
P21932	Adenylate cyclase type 3 (ADCY3)	+2.23
P29995	Inositol 1,4,5-trisphosphate receptor type 2 (ITPR2)	+2.07
Q62940	E3 ubiquitin-protein ligase (NEDD4)	+2.01
P17046	Lysosome-associated membrane glycoprotein 2 (LAMP2)	+1.91
G3B674	Claudin domain containing 1, isoform CRA_b (CLDND1)	+1.89
A0A0U1RRT8	Septin (SEPT6)	−2.24
P63025	Vesicle-associated membrane protein 3 (VAMP3)	−2.30
G3V852	RCG55135, isoform CRA_b (TLN1)	−2.32
Q7M0E3	Destrin (DSTN)	−2.58
G3V7Q6	Proteasome subunit beta (PSMB5)	−2.74
P68511	14-3-3 protein eta (YWHAH)	−3.27
Q920Q0	Paralemmin-1 (PALM)	−5.43

## Data Availability

Data are contained within the article and additional data in the Appendix A.

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
