# Peer review of "Identification of YWHAH as a Novel Brain-Derived Extracellular Vesicle Marker Post Long-Term Midazolam Exposure during Early Development"

_cells, 2023, doi:10.3390/cells12060966_

Round 1

Reviewer 1 Report

The authors reviewed “Identification of YWHAH as a Novel Brain-Derived 2 Extracellular Vesicle Marker Post Long-Term Midazolam 3 Exposure during Early Development.” Overall, the work is interesting and will be beneficial for the fellow researchers. I will recommend it after some minor revisions.

1.      Which are the formulations analyzed by TEM named are lyophilized or suspended in water.

2.      “There were several potential proteins in the list that we aimed to validate; however, the results were insignificant. One hit we successfully identified and validated was 14-3-3 eta protein (YWHAH), an adapter protein responsible for intracellular signaling and regulatory pathways.” Did you validated all protein?

The above mentioned points can be thought over and justified by authors, minor revision would be needed to go ahead.

Author Response

The authors reviewed “Identification of YWHAH as a Novel Brain-Derived 2 Extracellular Vesicle Marker Post Long-Term Midazolam 3 Exposure during Early Development.” Overall, the work is interesting and will be beneficial for the fellow researchers. I will recommend it after some minor revisions.

  1. Which are the formulations analyzed by TEM named are lyophilized or suspended in water.

- Thank you very much for taking the time to review our manuscript. We have included the information about the TEM buffer in lines 142-145 in the revised manuscript. Briefly, post BDEV isolation, 90 mL TEM ( 2% glutaraldehyde, 2% paraformaldehyde, and 0.1 M phosphate buffer) was added to 10ml of the BDEV aliquot and further analyzed by TEM at the UNMC electron microscopy core.

  1. “There were several potential proteins in the list that we aimed to validate; however, the results were insignificant. One hit we successfully identified and validated was 14-3-3 eta protein (YWHAH), an adapter protein responsible for intracellular signaling and regulatory pathways.” Did you validate all proteins?

- We apologize for the way the sentence was structured and have now revised it to reflect our findings (Line 323) better. We meant to convey that validation of hits identified from a high throughput screen is vital to further confirm their expression levels, given the risk of false positives. Also, from a biological perspective, it is common practice that key hits are picked based on their novelty posed in the context for post-validation. We employed a similar approach given the lack of information on the role of YWHAH in long-term midazolam exposure on neurodevelopment, thus validating this in our study.

Reviewer 2 Report

In the manuscript “Identification of YWHAH as a Novel Brain-Derived Extracellular Vesicle Marker Post Long-Term Midazolam Exposure During Early Development” authors isolated Brain-Derived Extracellular Vesicle (BDEV) from mice treated with Midazolam and characterized the BDEV using proteomics. The manuscript should address the following concerns which makes it unsuitable for publication.

1.     How did the authors determine 1mg/kg dose of MDZ?

2.     Midazolam treatment plan is not clear. How much higher dose is given and in what interval?

3.     Please provide loading control for western blots. Can authors explain why CD63 was not detected in brain lysates? Without that data cannot be interpreted.

4.     In Fig 1C, what is the n for the pups for control and MDZ treatment?

5.     Authors should add the p values for graphs.

6.     Authors should provide the primer list.

7.     At what time point RT-PCR for ESCRT pathway genes were carried out?

8.     Naming in Table1 and Figure 4 should match. Please correct it.

9.     Can the authors explain how gene ontology was performed? Did they include non-significant proteins?  Figs 5 and 6 are not clear to me. Whether these proteins interacted with each other?  

1.   Section 3.5 Why did the hits identified could not be validated? This creates doubt on validity of experiments.

1.   Ponceau S staining is not visible. Authors should upload a loading control for WB.

Author Response

Reviewer 2.

In the manuscript “Identification of YWHAH as a Novel Brain-Derived Extracellular Vesicle Marker Post Long-Term Midazolam Exposure During Early Development” authors isolated Brain-Derived Extracellular Vesicle (BDEV) from mice treated with Midazolam and characterized the BDEV using proteomics. The manuscript should address the following concerns which makes it unsuitable for publication.

  1. How did the authors determine 1mg/kg dose of MDZ? Midazolam treatment plan is not clear. How much higher dose is given and in what interval?

- Thank you very much for taking the time to review our manuscript and citing some critical inputs to help strengthen the manuscript. The rationale for our dosage regimen was to closely mimic the practice employed in the NICU on the babies. Since MDZ induces strong sedation, we were very cautious in initiating our regimen at a low dose (cf. 1mg/kg induced mild sedation) and the subsequent highest dose (10mg/kg on day 6 onwards until weaning at 21days induced moderate sedation). This regimen followed in our study has been published in our previous study (Nguyen et al. IJMS, Ref # 3) and the pertinent information are included in the methodology section (lines 81-90).

  1. Please provide loading control for western blots. Can authors explain why CD63 was not detected in brain lysates? Without that data cannot be interpreted.

- We have parsed through several published papers that majorly point to the issue of a house keeping protein for western normalization of EV samples. We also unfortunately had endured a similar issue as the expression levels of key housekeeping proteins, such as GAPDH and beta-actin were very erroneously expressed. Also, ISEV guidelines state that equal amount of tissue from the control and experimental samples are important including the same amount of protein loaded for subsequent interpretation of the data as acceptable. While we have used same starting weight of the tissue from the control and treatment groups keeping in mind the guidelines, the only way to assess equal loading is via staining with ponceau S and thus employed this approach. Pertaining the expression of CD63 in the brain lysates, we kindly refer the reviewer to the supplemental and under figure 1A that shows its expression in the brain lysate samples (with an asterisk) .

  1. In Fig 1C, what is the n for the pups for control and MDZ treatment?

- This information has now been included.

  1. Authors should add the p values for graphs.

- This information has now been included.

  1. Authors should provide the primer list.

- We used a customized array plate from applied biosystems that has the pertinent pre-designed primer/probes by the company. We have included all the relevant information on the accession numbers of the genes and their respective ID in the new supplemental data (Table S4) in the revised manuscript.

  1. At what point was RT-PCR for ESCRT pathway genes carried out?

- The RT-PCR for ESCRT pathways genes was carried out on postnatal 21 days old brains from the control and treatment groups.  

  1. The naming in Table 1 and Figure 4 should match. Please correct it.

- Thank you for pointing this out. This is now corrected.

  1. Can the authors explain how gene ontology was performed? Did they include non-significant proteins? Figs 5 and 6 are not clear to me. Whether these proteins interact with each other?

- The ClueGO analysis, as indicated in lines 195-197, is done on the differentially expressed proteins (DEPs) which are significant. For Figures 5B and 6B, GO terms are represented as nodes, and the node size represents the term enrichment significance; the proteins (red labeled) are the ones that are associated with those GO terms. Some proteins are associated with 1-2 common GO terms; for instance, Ywhah and Nedd4 in figure 6B are associated with sodium channel regulator activity and regulation of sodium ion transmembrane transporter activity.

  1. Section 3.5 Why did the hits identified could not be validated? This creates doubt about the validity of experiments

- Please refer to point #2 under reviewer 1.

  1. Ponceau S staining is not visible. Authors should upload a loading control for WB.

- Please refer to point # 2 above.